# Supramolecular coupling of cylindrical micelles following seeded-growth

Wenhao Gao[1,6], Kaiwen Sun[1,6], Xiaosong Wang [2] ✉, Liang Gao [1,3] ✉, Jiaping Lin [1,3,4,5] ✉, Chengyan Zhang[1] & Chunhua Cai [1,3]

Macromolecular coupling is a widely used technique for industrial materials, while supramolecular coupling is ubiquitous in biological systems. Although the designed synthesis of one-dimensional self-assembled nanostructures via crystallization-driven self-assembly and liquid-crystallization-driven self-assembly has been realized, end-to-end coupling of cylindrical micelles is rarely reported. Unlike crystallization, liquid-crystallization features fluidity under certain conditions. The cylindrical micelles prepared via liquid-crystallization-driven self-assembly possess less organized liquid crystalline blocks at the two partially open ends, originating the end-to-end coupling to lower the free energy. The interaction strength of solvents with liquid crystalline blocks is a pivotal parameter that can be used to switch on or off the coupling. Theoretical simulation is consistent with experimental work, supporting the mechanism. The supramolecular coupling offers opportunities for designing complex polymeric liquid crystalline nanostructures.

Covalent coupling refers to the intermolecular reaction of macromolecules, a technique widely used for industrial materials[1–3]. On the other hand, supramolecular coupling, though rarely reported, is ubiquitous in natural systems[4]. The spontaneous coupling of collagen fibers is one of the essential supramolecular events for their functions[5–9]. However, the mechanisms underlying this coupling remain largely unknown and are yet to be explored to empower supramolecular synthesis.

Self-assembly of block copolymers has been developed for controlled synthesis of nanostructures with a high degree of complexity and enhanced functions via crystallization-driven self-assembly (CDSA)[10–16]. In CDSA, micelles with crystalline cores exposed to solvents serve as nucleation sites, which induce the crystallization of a second crystalline block copolymer, generating block co-micelles with defined chemical compositions and morphologies[17–22]. Notably, the epitaxial growth has also been observed for block copolymers with a liquid-crystallization (LC) block[23–34]. The fluidity of liquid crystals, unlike the static state of the crystalline core, allows the rearrangement

and ordering of LC blocks within the micelle cores[35–38]. This dynamic behavior provides additional pathways for micelle growth and structure design, such as aggregation-fusion growth[24,25,35]. In contrast to the epitaxial growth of crystalline copolymers, under certain conditions, the LC block copolymers can form small aggregates first, and then fuse with the LC cores of seeds for growth. The combination of the solvophobicity of LC cores and the subsequent rearrangement of LC blocks drives growth via the fusion of small aggregates[25,36,37], resulting in cylindrical micelles with defined lengths based on the number of seeds used. The fusion can be adjusted by varying the solvent-LC block interactions, as the solubility of LC blocks is a pivotal parameter for the growth[24,29,39–41]. It is, therefore, possible to drive LC ends to meet, collide, and fuse, after the growth, to realize end-to-end coupling of grown cylinders into nanowires. Each nanowire will contain more than one seed.

Herein, we report the end-to-end coupling of (PBLG-*b*-PNIPAM) (PBLG: poly(γ-benzyl L-glutamate), PNIPAM: poly(N-isopropyl acrylamide)) long cylindrical micelles, forming elongated nanowires. The

[1]School of Materials Science and Engineering, East China University of Science and Technology, Shanghai, China. [2]Department of Chemistry, Waterloo Institute for Nanotechnology, University of Waterloo, Waterloo, Canada. [3]Shanghai Key Laboratory of Advanced Polymeric Materials, East China University of Science and Technology, Shanghai, China. [4]Key Laboratory for Ultrafine Materials of Ministry of Education, East China University of Science and Technology, Shanghai, China. [5]Frontiers Science Center for Materiobiology and Dynamic Chemistry, East China University of Science and Technology, Shanghai, China. [6]These authors contributed equally: Wenhao Gao, Kaiwen Sun. ✉e-mail: xiaosong.wang@uwaterloo.ca; lianggao@ecust.edu.cn; jlin@ecust.edu.cn

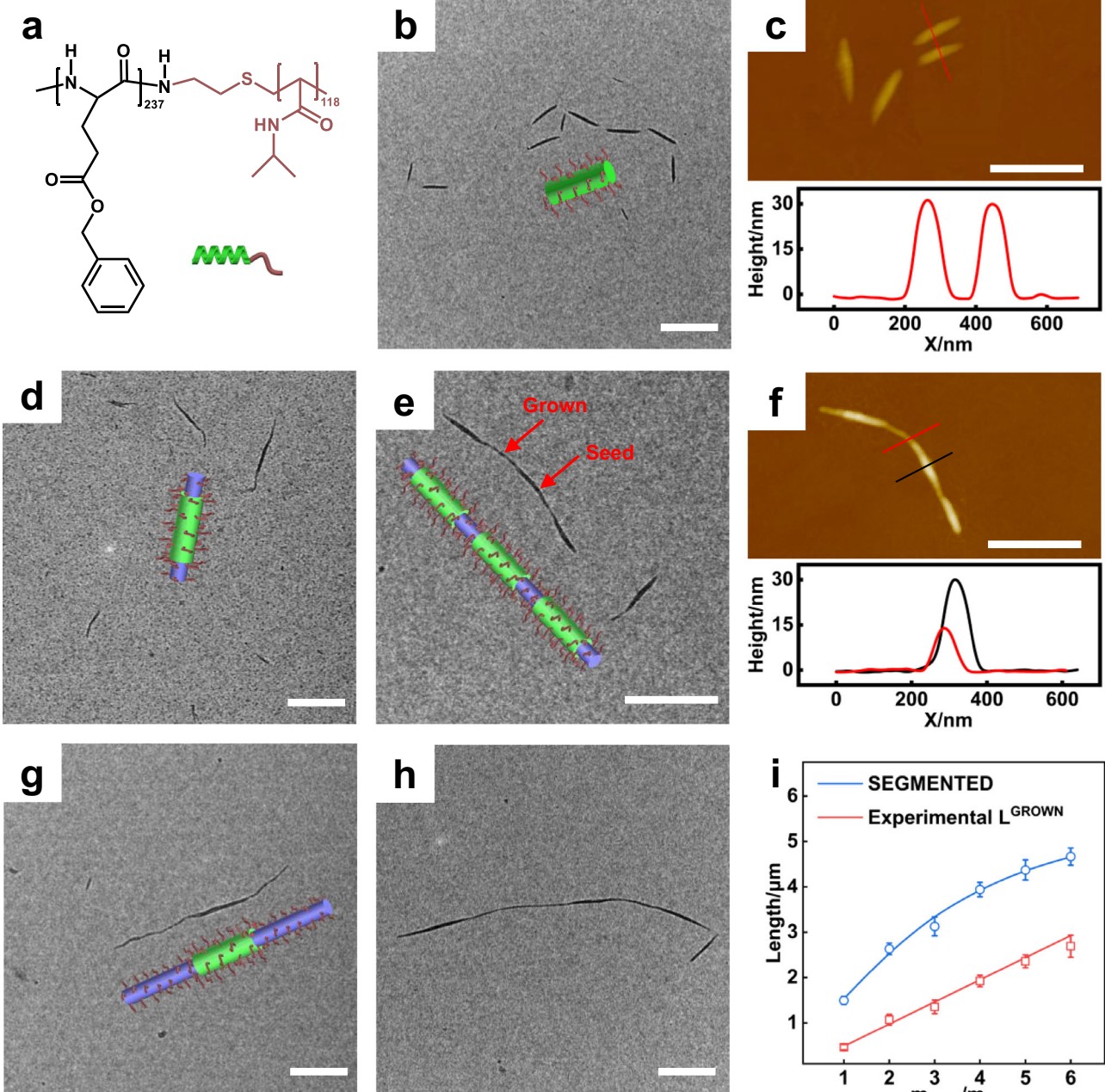

**Fig. 1 | Formation of segmented nanowires via end-to-end coupling. a** Chemical structure and scheme of PBLG$_{237}$-*b*-PNIPM$_{118}$ block copolymers. **b** TEM and **c** AFM images of the initial cylindrical micelles, *i.e.*, seeds, formed by PBLG$_{237}$-*b*-PNIPAM$_{118}$, where THF/MeOH is 45.5/54.5 in volume. **d** TEM image of the grown cylindrical micelles after incubating them for 6 h. **e** TEM and **f** AFM images of the segmented hierarchical nanowires after incubating for 48 h. For the seed solution after adding copolymers, THF/MeOH/DMF is 45/54/1 in volume. The insets in figures (**c**) and (**f**) are the AFM height profiles along the selected regions. **g, h** TEM images of the assemblies after incubating for **g** 6 h and **h** 48 h when the feed ratio is 6.0. **i** The variations of the total length of the segmented nanowires and the length L$^{GROWN}$ of the total grown parts with the unimer-to-seed ratio after incubating for 72 h. Error bars represent mean ± standard deviation, $n \geq 200$. Scale bars: 1 μm.

coupling occurs following the aggregation-growth in solvent systems. As indicated in the theoretical simulation, the LC blocks are not perfectly organized into cholesteric liquid crystals at the ends of cylindrical micelles. The coupling enables these LC blocks to be reorganized and align with the micelles' internally ordered structures, ultimately stabilizing the linkage. Theoretical simulations indicate that the chain rearrangement process in the coupling, relative to the fusion, involves a larger number of liquid crystal blocks and structural adjustments in both connected cylindrical micelles. This work illustrates the unique capability of liquid-crystallization-driven self-assembly (LCDSA), which complements CDSA and offers possibilities for designing and constructing sophisticated nanostructures with intricate hierarchical features.

## Results

### Formation of segmented nanowires via end-to-end coupling

Poly(γ-benzyl L-glutamate) (PBLG) is a hydrophobic, rod-like polypeptide with a rigid α-helical conformation. When copolymerized with a hydrophilic coil-like block, such as poly(N-isopropyl acrylamide) (PNIPAM), the resulting rod-coil block copolymers can self-assemble into micelles in selective solvents for the hydrophilic block. Here, PBLG$_{237}$-*b*-PNIPAM$_{118}$ (Fig. 1a, subscripts indicate the degree of polymerization) was used for the selective precipitation in tetrahydrofuran/methanol = 45.5/54.5 (v/v), in which tetrahydrofuran (THF) is a common solvent and methanol (MeOH) is a selective solvent. The details of polymer synthesis and micelle preparation are provided in

Section 1.1 and 1.2 of the Supplementary Information (SI). The rationale for choosing the PBLG$_{237}$-$b$-PNIPAM$_{118}$ system is given in Section 1.7 of the SI. Transmission electron microscopy (TEM, Fig. 1b) and atomic force microscopy (AFM, Fig. 1c) confirmed the cylindrical morphology. The average length is ~490 nm, and the polydispersity index is 1.10 (Supplementary Fig. 4). The AFM height profile (Fig. 1c, inset) showed a micelle height of ~30 nm. These cylindrical micelles were used as seeds for subsequent seeded growth experiments.

A solution of PBLG$_{237}$-$b$-PNIPAM$_{118}$ in $N$, $N'$-Dimethyl Formamide (DMF, 2.0 g/L, 40 μL) was added to the seed micelle solution, adjusting the solvent composition to THF/MeOH/DMF = 45/54/1 (v/v/v). DMF, relative to THF, has a better solvation power for PBLG-$b$-PNIPAM. For seeded growth, we need to prepare a highly concentrated polymer solution to minimize the amount of good solvents in the growth system, so DMF is used. The small amounts of DMF do not affect either micelle growth or coupling; instead, it dissolves well the added copolymer during feeding (Section 1.2 of the SI). When the mixture was incubated for 6 h, the length of cylindrical micelles increased from ~490 nm to ~1200 nm, as indicated by the TEM image. As shown in the image, the center segment has a broader width than the two flanks (Fig. 1d). This suggests that the growth occurs at two ends of the seeds, and the LC blocks are organized into a narrower LC in the grown segment. Thinner grown segments are likely due to the lower ordering of LC blocks resulting from the growth process in different solvent conditions. After 48 h, nanowires with lengths up to ~2600 nm are formed and the thin segments of ~15 nm and the broad segments of ~30 nm alternate along the nanowires as indicated in TEM images (Fig. 1e). The AFM confirms it and reveals the alternative arrangement of the heights of ~15 and ~30 nm (Fig. 1f). It indicates that the segmented nanowires result from the end-to-end coupling of the cylindrical micelles. In addition, based on the difference in width (Supplementary Fig. 5), we can quantitatively analyze these TEM images using the Image-Pro Plus software and automatically identify the seed and grown segments per nanowire.

Based on the experimental results, nanowire formation occurs in two stages: fusion growth and end-to-end coupling. We investigated how varying the unimer-to-seed feed ratio affects this process. To achieve a high feed ratio, we diluted the initial seed solution with the same mixed solvent (THF/methanol = 45.5/54.5, v/v) and then added the copolymer solution (experimental details in Section 1.8 of the SI). As shown in Fig. 1g, Supplementary Fig. 7a, when the feed ratios are 2.0 and 6.0, incubated for 6 h, the lengths of cylindrical micelles are ~1150 nm (Supplementary Fig. 7a) and ~2700 nm (Fig. 1g), respectively. As expected, the larger the feed ratio, the longer the cylinders. After 48 h, the segmented nanowires (SEGMENTED) formed regardless of feed ratios (Fig. 1h, Supplementary Fig. 7b). We analyzed the seed (SEED) and grown (GROWN) segments in nanowires, which are distinguished from TEM images. The thick and thin segments in the images represent SEED and GROWN. By analyzing over 200 TEM images using Supplementary Equations (4, 5), we calculated the average number of SEED per nanowire and the contribution of the GROWN segments to the total length for SEGMENTED.

As shown in Fig. 1i, the total length for GROWN (L$^{GROWN}$) increases linearly with the feed ratio, confirming the complete fusion of added copolymers to seed ends. However, the total length for SEGMENTED does not linearly follow the feed ratios. When the ratio is over 3, the growth of SEGMENTED is retarded (blue line, Fig. 1i). This is due to the less number of SEED (N$^{SEED}$) in the SEGMENTED prepared by large feed ratios (Supplementary Fig. 7d), suggesting less end-to-end coupling. It is reasonable, as a longer GROWN block, resulting from the larger feed ratios, reduced the mobility and thus hindered end-to-end coupling.

Time-dependent dynamic light scattering (DLS) curves for the solution of seeds (in THF/MeOH = 45.5/54.5, v/v), unimers of the block copolymer (in THF/MeOH/DMF = 45/54/1, v/v/v), and the solution that contains seeds and unimers (in THF/MeOH/DMF = 45/54/1, v/v/v) are

illustrated in Fig. 2a–c, respectively. In the solution of seeds, the $R_h$ remained unchanged for the first 24 h, followed by a slight increase after 48 h (Fig. 2a), suggesting some end-to-end coupling of seeds occurred. The $R_h$ measured from the solution of unimers is *ca.* 20 nm and remained unchanged over time (Fig. 2b, Supplementary Fig. 9a, b), so the unimers self-aggregated into small aggregates that do not fuse. AFM characterization of these aggregates shows that the height of the small aggregate is *ca.* 15 nm (Supplementary Fig. 9c, d), which is less than that of the seeds (*ca.* 30 nm). The difference between the two solutions is the solvents, which appear to be a factor affecting the dimensions of the assemblies.

The DLS curve shows two $R_h$ of 15 nm and 50 nm for the solutions of seeds in the presence of added unimers (Fig. 2c), attributed to the aggregates of added unimers and seeds. Within the first 24 h, the signal due to small aggregates becomes weaker, and the larger one increases to 56 nm, indicating the occurrence of growth. As aggregates do not fuse, the growth is the fusion of aggregates with seeds (aggregate-seed fusion). At 24 h, the small aggregates are consumed completely, so no aggregate-seed fusion continues. However, at 48 h, a DLS curve with a tail at larger $R_h$ values is found, and the average $R_h$ shifts to larger values at 72 h. This increase in $R_h$ is attributed to end-to-end coupling. The TEM images corresponding to various time points in the DLS curves (Supplementary Fig. 10) further support the fusion growth between seed and small aggregates and the following coupling behavior of the grown micelles. The temporal distributions of N$^{SEED}$ and L$^{GROWN}$ for the 54.0 vol% MeOH condition (Supplementary Fig. 11) indicate that nearly all initial seeds have participated in coupling. The sequential aggregate-seed fusion growth and end-to-end coupling are explained by the mobility of the aggregates and the growing cylinders. The fusion occurs first because it is a relatively faster process. It is attributed to the faster mobility of the small aggregates and the higher possibility for the aggregates to collide with the ends of the seeds. The coupling of two longer cylinders with slower mobility occurs following the fusion process.

As a typical LC polymer, the PBLG adopts an ordered packing structure within the micellar core, resembling a cholesteric LC phase[42,43]. Synchrotron radiation wide-angle X-ray scattering (WAXS) was performed to investigate these LC structures. Samples were dried from the solutions for the measurement. The WAXS diffraction pattern of seed micelles (Fig. 2d) showed multiple diffraction rings, indicating ordered nanostructures. Figure 2e compares the WAXS spectra for seeds, small aggregates of the unimers, and grown micelles. The peak at $q = 4.4$ nm$^{-1}$ ($d = 1.43$ nm) reflected the packing distance between parallel PBLG chains in the micelle core. We also performed using a solution sample, a peak at $q = 4.4$ nm$^{-1}$ is also revealed (Fig. 2f), consistent with the measurement using dried samples. Therefore, the LC structures in the solution and after drying are the same. The aggregates have a weaker peak intensity at $q = 4.4$ nm$^{-1}$, suggesting a low chain ordering. Upon growth, the signal is intensified, and diffraction patterns form like seeds. It indicates that the fusion growth is driven by LC. PBLG blocks in small aggregates adjust their orientation to match the seed micelles, facilitating the formation of long cylindrical micelles to lower the free energy.

The WAXS results confirmed that ordered packing and chain rearrangement of PBLG blocks drive the growth of cylindrical micelles. To further investigate this process, theoretical simulations were conducted using a coarse-grained model (R$_6$C$_3$), where R$_6$ and C$_3$ represent the PBLG and PNIPAM blocks, respectively. The interaction strength ($\varepsilon_{RR}$) in the Lennard-Jones potential controlled PBLG hydrophobicity, with $\varepsilon_{RR} = 2.7\varepsilon$ ($\varepsilon$ is the energy unit in the simulation) used to simulate micelle formation (simulation details in Section 2.1 and 2.2 of the SI). The cylindrical seed micelles were first simulated, followed by adding R$_6$C$_3$ copolymers.

At early stages (Fig. 3a, b), the unimers formed small aggregates, some of which fused with seeds. Over time (Fig. 3c), more aggregates

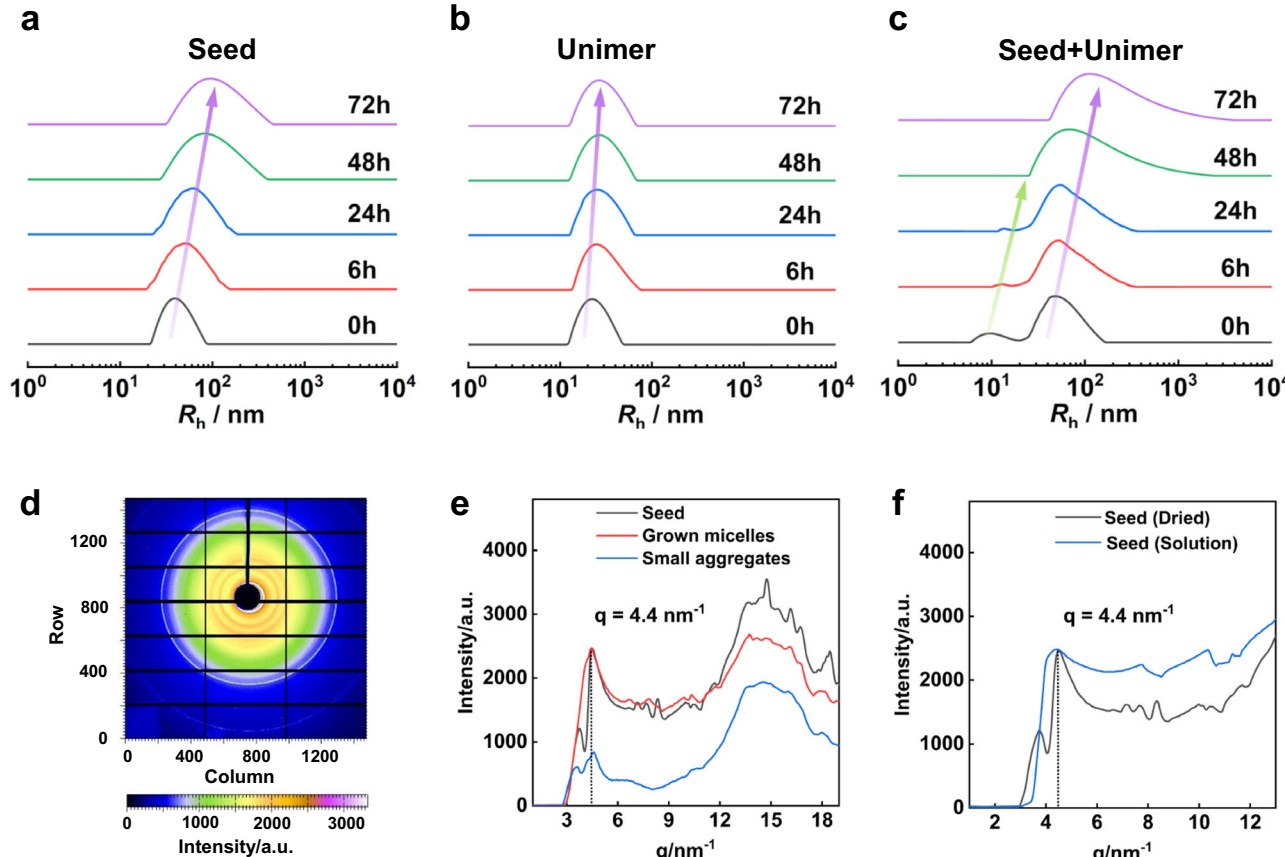

**Fig. 2 | Micelle growth and coupling process via liquid-crystallization-driven self-assembly. a–c** Temporal evolution of the $R_h$ of the **a** seed solution (in THF/MeOH = 45.5/54.5, v/v), **b** unimers solution (in THF/MeOH/DMF = 45/54/1, v/v/v), **c** seed solution containing added copolymer unimers (in THF/MeOH/54/

1, v/v/v). **d** WAXS pattern of the seed micelles sample. **e** WAXS spectra of the micelle samples (seed micelles, grown micelles, and small aggregates). **f** WAXS spectra of the dried sample and the solution sample of micelle seeds.

fused to the cylinders grown from the seeds, aligning with experimental findings. Notably, end-to-end coupling was absent at this stage. However, as the incubation prolongs to $1.0 \times 10^5 \tau$ ($\tau$ is the time unit in the simulation), coupling occurs, forming segmented nanowires (Fig. 3d). Both experiments and simulation indicate that the formation of segmented nanowires involves two sequential processes (Fig. 3i, Supplementary Movie). It starts with **Fusion Growth** – Small aggregates, which do not have a perfectly ordered LC core, attach to seed ends and adjust their chain alignment to match the cholesteric LC structure of the seeds (Supplementary Fig. 21). This restructuring stabilizes micelle growth. Following this process, **End-to-End Coupling** occurs—As grown micelles collide, their partially exposed ends act as active sites for coupling (Fig. 3e). The rod blocks in the core at the coupling junction show weak alignment right after the coupling (Fig. 3f). Over time, their orientation adjusts (Fig. 3g, h), resulting in a continuous cholesteric LC structure. This LC-driven self-assembly relies on chain fluidity and rearrangement, though full alignment requires extended time due to the twisting rearrangement of the coupled LC blocks. The fusion growth between a seed and a small aggregate takes *ca.* $1500\,\tau$ for the chains to rearrange, while the coupling of two grown cylindrical micelles takes about $5500\,\tau$ (Supplementary Fig. 22).

## Solvent effects on micelle growth and coupling

The quality of solvents influences the mobility of LC blocks, thereby affecting micelle growth and coupling. In a THF/MeOH mixture, the methanol content significantly impacts micelle formation. As shown in Fig. 4a, b, increasing the methanol content from 64.3 vol% to 70.6 vol%

reduces $N^{SEED}$ in nanowires. At 70.6 vol%, SEGMENTED is minimal, and growth is primarily driven by fusion at seed ends, resulting in long cylindrical micelles (Fig. 4b). As measured from more than 100 samples in TEM images, the SEGMENTED mainly exhibits a seed number of 1 (Fig. 4b, inset). The heights of the grown parts or seed parts remained nearly unchanged with increasing the methanol content (Supplementary Fig. 12).

To quantify this effect, we analyzed the samples taken at specific intervals during the growth in the systems with various methanol contents from 54.0 vol% to 70.6 vol%. For the system with 54.0 vol% methanol, $L^{GROWN}$ grows rapidly to 750 nm in 6 h and levels off at ~1100 nm after 48 h. In the first few hours, $N^{SEED}$ is less than 1.5, suggesting that the early-stage growth (within 6 h) is dominated by fusion growth. Following the growth, the coupling becomes significant, and the $N^{SEED}$ increases to 3.3 at 48 h. In contrast, when methanol is 70.6 vol%, the final $N^{SEED}$ and $L^{GROWN}$ are only 1.7 (Fig. 4c) and 550 nm (Fig. 4d), respectively, after 48 h, indicating that both fusion and coupling are retarded at high methanol content. For the other methanol content condition, see Supplementary Fig. 13. The obtained results suggest that a higher methanol content reduces the mobility of LC blocks. In the simulations, the enhancement of the $\varepsilon_{RR}$ (higher methanol content) makes the chain rearrangement more difficult and limits the fusion growth and coupling of micelles (Supplementary Fig. 23), which is consistent with the experimental observation.

To explore solvent effects further, we prepared seed micelles in different initial solvents or selective solvents. In the first case, the common solvent THF was replaced by dioxane (Diox), while in the latter case, the selective solvent methanol was replaced by ethanol

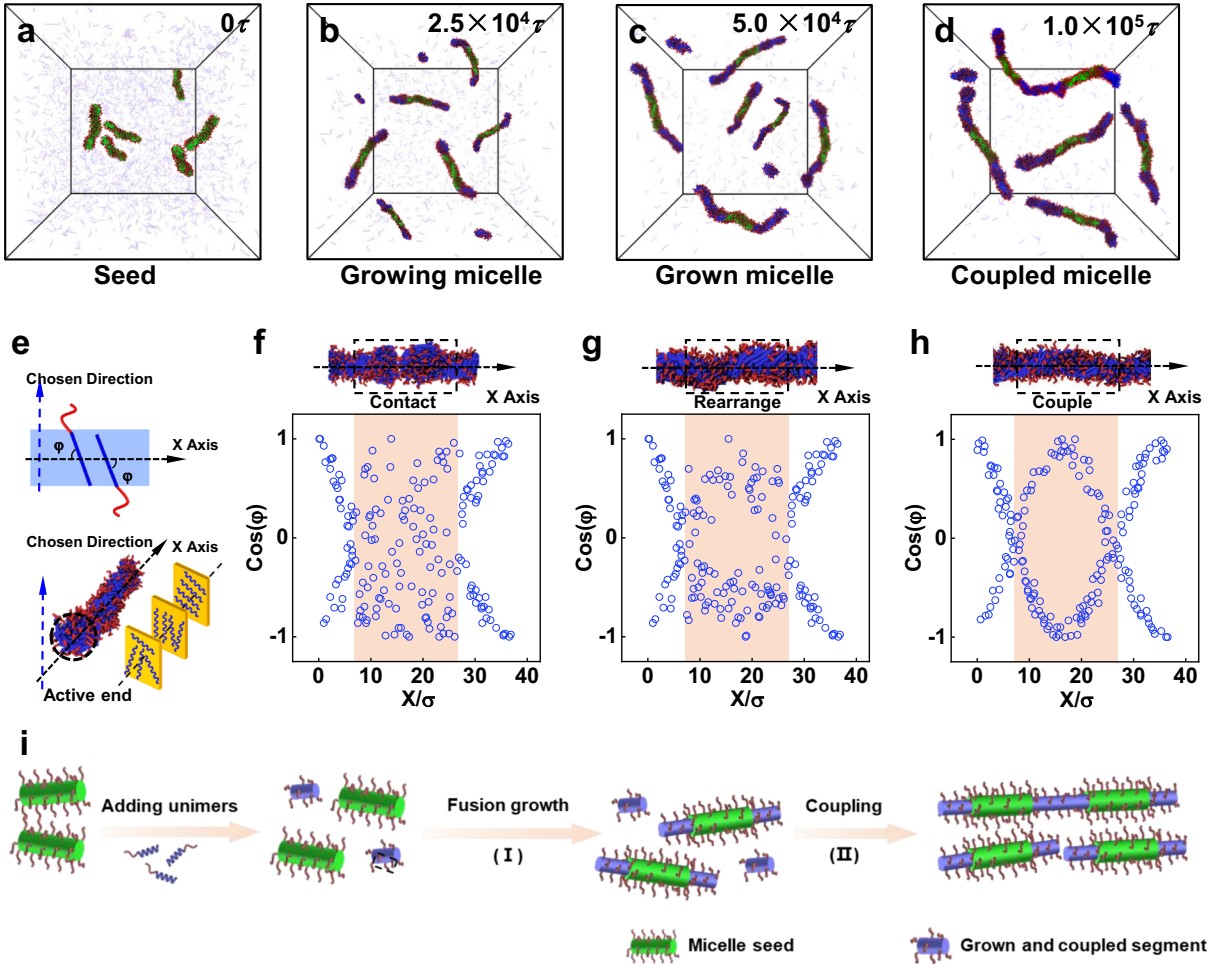

**Fig. 3 | Formation mechanism of segmented nanowires revealed by simulations. a–d** Simulation snapshots of aggregates after feeding unimers at the simulation time of **a** $0\,\tau$, **b** $2.5 \times 10^4\,\tau$, **c** $5.0 \times 10^4\,\tau$, and **d** $1.0 \times 10^5\,\tau$, respectively. For clarity, the rod blocks in the seeds and the added unimers are colored green and blue, respectively, and the free unimers are hyalinized. **e** Typical structure of the newly grown part and the definition of the orientation angles, $\varphi$, in the micelle core. **f–h** Variations in the cosine of $\varphi$ along the X-axis of the cylindrical micelles. Only the two ends of the grown micelles and their coupled region are presented in the figures. **i** Schematic illustration of (I) Fusion growth and (II) Coupling behaviors via LCDSA, where green and blue objects represent the micelle seed, the grown and coupled segment with different degrees of LC ordering.

(EtOH). The obtained seeds have lengths of ~385 nm and ~760 nm, respectively (Supplementary Figs. 14b, 14d). Then, DMF solutions of the block copolymer (2.0 g/L, 40 µL) were added to these seed solutions for growth.

The growth in the resulting Diox/MeOH/DMF system leads to cylindrical micelles with a length of ~1200 nm (Fig. 5a, Supplementary Fig. 15a). Figure 5a inset and Fig. 5d (blue line) show that most cylinders have an $N^{SEED}$ of 1.0, suggesting coupling is suppressed when THF is replaced by dioxane. It is attributed to the better solvent-LC block interaction[24,29,44,45], which helps the orderings of PBLG blocks in micelle cores (Supplementary Fig. 16), and thus reduces their possibility for subsequent coupling, which needs space for chain rearrangements.

In contrast, the THF/EtOH/DMF system facilitates the formation of SEGMENTED (Fig. 5b), in which the $N^{SEED}$ reaches 4.0 (Fig. 5b inset, and Fig. 5d, green line). In addition, the $L^{GROWN}$ reaches ~3000 nm (Fig. 5b, Supplementary Fig. 15b), suggesting enhanced fusion growth as well. Since the interaction between the selective solvent ethanol and the LC block is relatively weaker[24,46,47], it could help both growth and coupling. Moreover, compared to the Diox/MeOH/DMF system, the $N^{SEED}$ in the Diox/EtOH/DMF system reaches 3.8 (Fig. 5c inset, and Fig. 5d, purple line). The seed micelles undergo effective growth and end-to-end coupling to form segmented nanowires, confirming that solvent

composition regulates chain rearrangement and thus the final nanowire morphology.

Such a solvent-dependent behavior originates from the LC ordering of the PBLG core-forming blocks. Because the end-to-end reactivity is dictated by how readily PBLG chains can rearrange at micelle edges, the solvation environment plays a decisive role. In selective alcohols, methanol solvates PBLG more strongly than ethanol. The weaker solvation of PBLG in ethanol introduces a mild solvophobic drive that facilitates chain mobility and LC reorganization. Consequently, growth and end-to-end coupling occur more efficiently in the THF/EtOH/DMF mixture.

The above experimental observations and simulation reveal the distinct features of LCDSA. Unlike CDSA, the driving force in LCDSA is the formation of LC structures, which do not have restricted crystalline plane matching, and more depends on the fluidity of the LC structure, allowing a dynamic chain rearrangement. This feature endows more kinetic pathways for seeded growth, which means more possible control in nanostructures. For example, due to fusion being mechanically related to the solubility of LC blocks, thereby the chain rearrangement ability can be adjusted by varying the solvent-LC block interactions[24,29]. This provides additional control and the potential for modulating the nanostructure. In the present work, a higher methanol

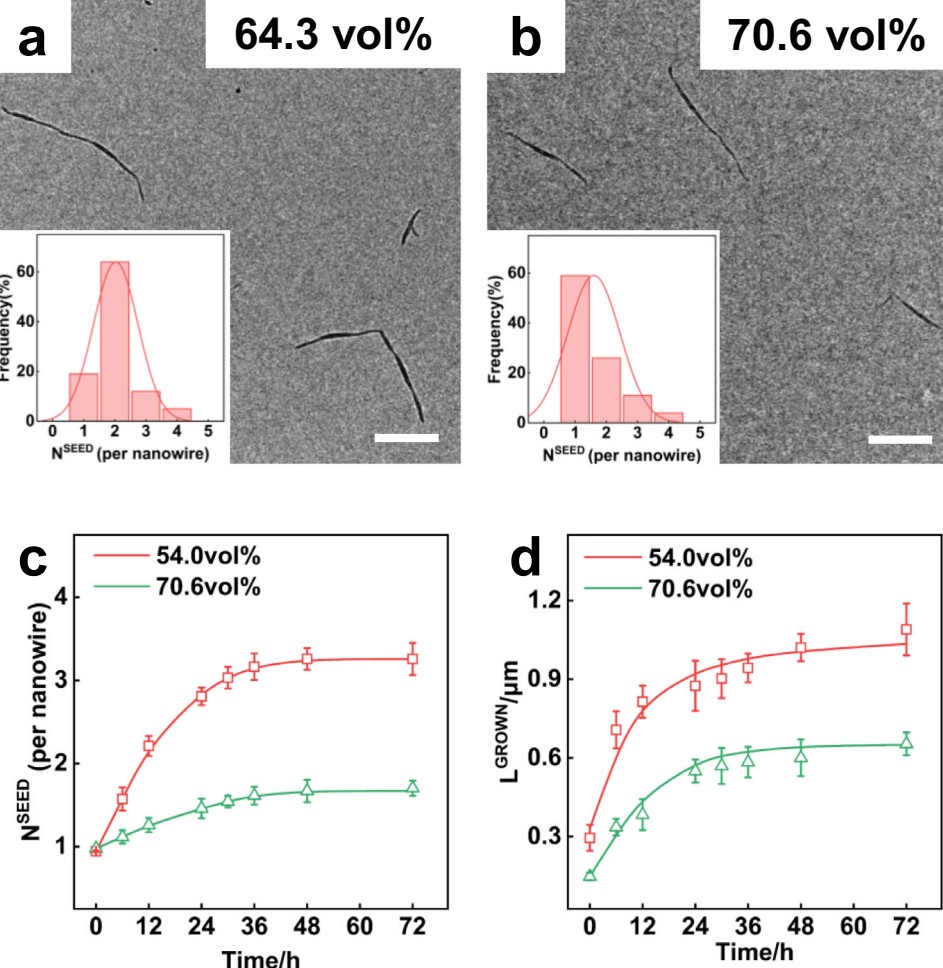

**Fig. 4 | Micelle end-to-end coupling regulated by the content of selective solvent. a, b** TEM images of the segmented nanowires after aging at 30 °C for 48 h at different methanol contents: **a** 64.3 vol%, **b** 70.6 vol%, the inset shows the seed number distribution per nanowire for different methanol contents. Scale bars: 1 μm. **c** Temporal variations of the seed number $N^{SEED}$ for the segmented nanowires formed at methanol contents of 54.0 vol% and 70.6 vol%. Error bars represent mean ± SD, $n \geq 200$. **d** Temporal variations of the length $L^{GROWN}$ of grown parts at methanol contents of 54.0 vol% and 70.6 vol%. Error bars represent mean ± SD, $n \geq 200$.

content increased the interactions between LC blocks, leading to reduced mobility of the LC blocks and retarding fusion growth. While the better common solvent (*e.g.*, dioxane) helps LC ordering and reduces the possibility of coupling. Conversely, in the selective solvent ethanol, the reduced interaction strength among LC blocks promotes coupling. By changing the solvent, we achieve the regulation of the number of seeds in the nanowires.

End-to-end coupling is not exclusive to LCDSA systems. Examples exist in amorphous and crystallizable polymers through hydrophobic interactions or crystallization-driven aggregation[32,39,48]. However, this work not only observed end-to-end coupling in LCDSA systems, but also uncovered how the LC nature of the PBLG core governs the fusion and coupling processes and, critically, how this LC-driven chemistry enables fine-tuning of coupling behavior. Thereby, the present work creates a hierarchically segmented nanowire architecture through a controllable end-to-end coupling mechanism driven by LC chain rearrangement. The segmented nanowires reported here feature alternating thin and thick segments along their long axis—an architecture fundamentally different from previous reported co-micelles or wormlike micelles[32,39,48]. The LC-based chain rearrangement is what enables fine control over coupling behavior; by tuning solvent composition, we can modulate this rearrangement and thus adjust the hierarchy of the segmented nanowires. Even when more seed solutions were multiple feedings, it still produced segmented nanowires, but

increased the average number of seeds per nanowire from ~3.3 to ~4.5 (Supplementary Fig. 8). This demonstrates that multiple feedings can adjust the hierarchical structure, yet the underlying assembly still proceeds through the exact LC-driven end-to-end coupling mechanism.

In addition, the present work also reveals the relationship between fusion growth and end-to-end coupling behaviors. It was found that when the growth contribution from coupling is excluded, the fusion growth of cylindrical micelles follows the principles of living polymerization. However, the coupling between cylindrical micelles requires more energy to accomplish the chain rearrangement, leading to the growth kinetics deviating from a linear relationship of living growth. This process resembles the termination reaction of two polymerizing polymer chains in free radical polymerization, providing insights for constructing controllable supramolecular reactions by utilizing the polymeric micelle subunits.

Finally, we want to clarify the generality of our strategy, that is, grow through fusion and then undergo LC-mediated end-to-end coupling. A key insight is that LC chain rearrangement—not simply micelle collision—governs whether controlled coupling can occur. Complementing experiments with theoretical simulations, the general rules for achieving controlled LC-driven coupling can be summarized as: 1) LC cores should rearrange more slowly during coupling than during fusion, requiring extended twisting and alignment. 2) The solubility

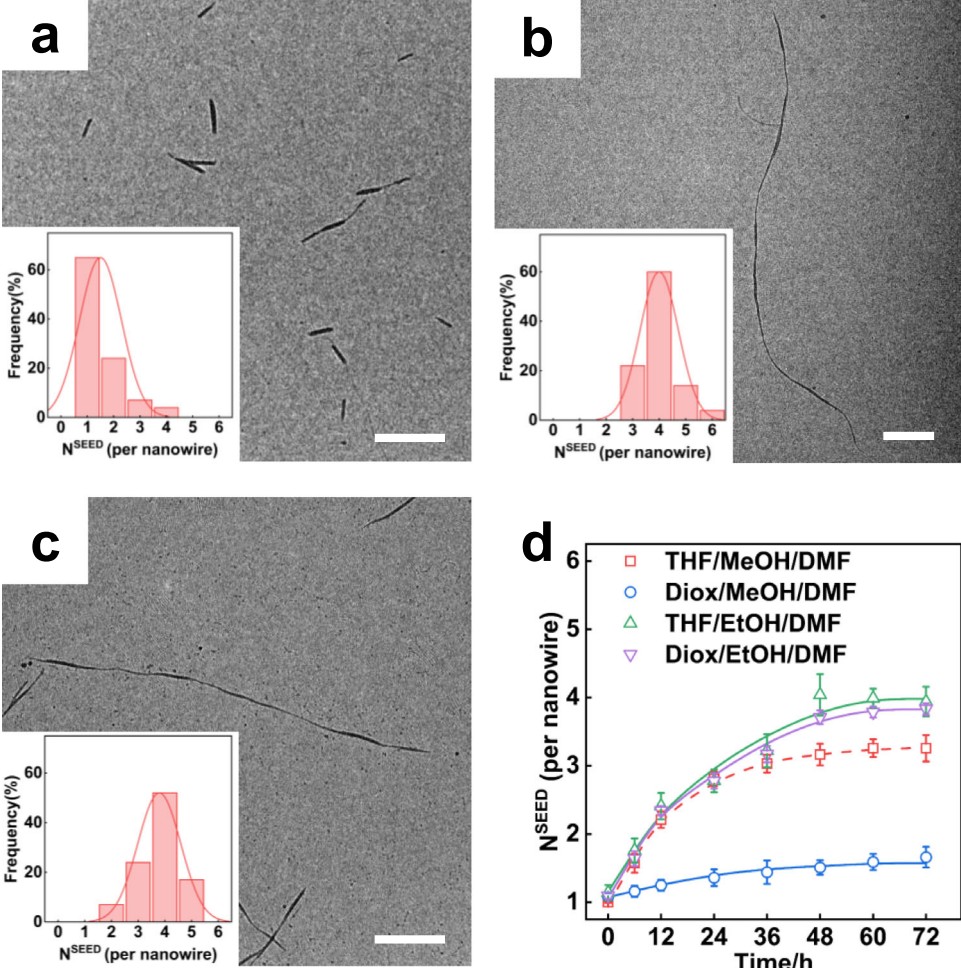

**Fig. 5 | Micelle end-to-end coupling regulated by the solvent composition.** **a**–**c** TEM images of the segmented nanowires prepared in **a** Diox/MeOH/DMF mixture solution, **b** THF/EtOH/DMF mixture solution, and **c** Diox/EtOH/DMF mixture solution, after aging at 30 °C for 48 h. The insets in (**a**–**c**) show the seed number distribution per nanowire. Scale bars: 1 μm. **d** Temporal variations of the seed number $N^{SEED}$ at different solvents. Error bars represent mean ± SD, $n \geq 200$.

parameters of the LC rod block and coil block should be sufficiently close to allow this rearrangement. 3) A moderate block ratio (*e.g.*, ~ 2:1 for PBLG$_{237}$-*b*-PNIPAM$_{118}$) is needed to balance packing frustration and flexibility. 4) Optimized solvent conditions should allow tunable LC ordering. When these conditions are met, the hierarchical segmented nanowires can form, demonstrating the mechanism's generality.

## Discussion

In summary, we developed segmented polymeric nanowires by controlling the fusion growth and end-to-end coupling of cylindrical micelles. Seed micelles with a cholesteric LC-like core were formed from PBLG-*b*-PNIPAM block copolymers. Upon adding LC copolymers and incubating, the micelles first grow through fusion and then couple end-to-end. Experiments and simulations revealed that the LC block arrangement plays a key role, with chain rearrangement regulated by solvent composition and type. During fusion growth, the PBLG blocks realign to match the LC-like seed core, while end-to-end coupling requires more rearrangements. This mechanism enhances understanding of micelle coupling via LCDSA and offers a precise strategy for designing polymeric hierarchical structures.

## Methods
### Materials
L-Glutamic acid γ-benzyl ester (BLG), ethyl acetate, petroleum ether, sodium chloride, and anhydrous magnesium sulfate were purchased

from Adamas-beta. Epoxypropane and triphosgene were purchased from TCI. *N*-isopropylacrylamide (NIPAM, 98%, TCI) was purified by recrystallization from n-hexane. 2,2-Azobis(2-Methylpropionitrile) (AIBN, 98%, TCI) was recrystallized from absolute methanol. 2-Aminoethanethiol Hydrochloride (AET·HCl, 95%, TCI) was used without further purification. Methanol (MeOH), Ethanol (EtOH), Tetra-hydrofuran (THF), *N*, *N'*-Dimethyl Formamide (DMF), and 1,4-Dioxane (Diox) of analytical grade were purchased from Adamas-beta. Deionized water was prepared in a Millipore Super-Q Plus Water System to a level of 18.2 MΩ cm resistance. The dialysis bag (14000 molecular weight cutoff) was provided by Serva Electrophoresis GmbH.

### Synthesis of PBLG-*b*-PNIPAM diblock copolymers
The monomer γ-Benzyl-L-glutamate-N-carboxyanhydride (BLG-NCA) was synthesized by the triphosgene method. Specifically, L-Glutamic acid γ-benzyl ester (BLG, 15 g), epoxypropane (20 mL), and triphosgene (11.7 g) were mixed in 225 mL of tetrahydrofuran. After stirring for 2 h at room temperature, the reaction was quenched with deionized water. The reaction mixture was then sequentially washed with ethyl acetate (150 mL) and saturated brine (200 mL). The organic phase was dried over anhydrous magnesium sulfate. Following the removal of the organic solvent under vacuum, the crude residue was purified by recrystallization in petroleum ether. The synthesis method of the initiator PNIPAM-NH$_2$ is as follows. AIBN was used as the initiator and AET·HCl as the chain transfer agent to synthesize monoamine-terminated PNIPAM (PNIPAM-

$NH_2$) by free radical polymerization of NIPAM in methanol at 60 °C. The PBLG-*b*-PNIPAM diblock copolymers were synthesized in anhydrous 1,4-dioxane by ring-opening polymerization of BLG-NCA monomers initiated by the PNIPAM-$NH_2$ macroinitiator. The reaction was performed in a flame-dried reaction eggplant flask under a dry nitrogen atmosphere at 15 °C for 3 days. After completion of the polymerization, the viscous reaction mixture was poured into anhydrous methanol. The resulting crude residue was dried under vacuum and then purified by recrystallization in methanol to obtain a white solid powder. More synthesis details are given in Section 1.1 of the SI.

## Characterizations of PBLG-*b*-PNIPAM diblock copolymers

The chemical structures and the degree of polymerization (DP) of the block copolymers were measured by the nuclear magnetic resonance spectrometer ($^1$H NMR, Avance 600 MHz, Bruker). The polydispersity (PDI) of block copolymers was determined by the gel permeation chromatography (GPC, PL GPC-50 plus, Varian) measurement. The calibration curve was obtained using narrow polydispersity PBLG polypeptide standards. The characterization details of PBLG-*b*-PNIPAM diblock copolymers are in Section 1.1 of the SI.

## Preparation procedure of seed micelles and segmented nanowires

The seed micelles were prepared through the selective precipitation method. Then, a solution of PBLG-*b*-PNIPAM copolymers dissolved in DMF (2.0 g/L, 40 μL) was added to the seed solution (4.4 mL, THF/Methanol, v/v, 45.5/54.5). The mixed solution was aged for various incubation times. All incubated solutions were dialyzed against deionized water for at least 3 days to ensure that the organic solvent had been removed before the characterization of aggregate structures. The self-assembly details are given in Section 1.2 of the SI.

## Characterizations of assemblies

The morphologies of assemblies were observed by transmission electron microscopy (TEM, JEM-1400, JEOL) operated at an accelerating voltage of 100 kV. The atomic force microscopy (AFM) measurements were performed with an XE-100 (Park Systems) by using the non-contact mode at room temperature in the air. Turbidity measurement was performed to determine the critical methanol content for the aggregation of block copolymers. The optical density (turbidity) was measured at a wavelength of 690 nm with a UV-vis spectrophotometer (UV-2550, Shimadzu). Synchrotron radiation Wide-angle X-ray scattering (WAXS) measurement was performed at beamline BL16B1 of the Shanghai Synchrotron Radiation Facility. The distance from the sample to the WAXS detector was calibrated to 192 mm, and the wavelength of the X-ray was 0.124 nm ($E = 10$ keV). A two-dimensional WAXS pattern was collected with a Mar165 CCD detector (2048 × 2048 pixels with a pixel size of 80 μm). The WAXS data were analyzed with the Fit2D software from the European Synchrotron Radiation Facility. The characterization details of aggregates are in Section 1.3 of the SI.

## Data availability

The data generated in this study are provided in the Supplementary Information/Source Data file. The raw data generated in this study have been deposited in the Figshare database https://doi.org/10.6084/m9.figshare.31146058. Additional data are available from the corresponding author upon request. Source data are provided with this paper.

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

## Acknowledgments

This work was supported by the National Natural Science Foundation of China (52394270, 52394271, 52573021). This work was carried out with the support of the Shanghai Synchrotron Radiation Facility, Beamline BL16B1 (2025-SSRF-PT-513613).

## Author contributions

L.G. and J.L. conceived the idea and supervised the project. W.G. designed and carried out the experiments. K.S. and C.Z. performed the simulations. W.G., X.W., L.G., and C.C. analyzed the data. W.G., X.W., L.G., and J.L. co-wrote the manuscript.

## Competing interests

The authors declare no competing interests.
