## [Transparent Peer Review file · Nature Communications]

Supramolecular coupling of cylindrical micelles following seeded-growth

Corresponding Author: Professor Jiaping Lin

Version 0:

Reviewer comments:

Reviewer #1

(Remarks to the Author)

This manuscript from Gao et al. presents a significant advancement in supramolecular polymer science by demonstrating controlled end-to-end coupling of cylindrical micelles via liquid-crystallization-driven self-assembly (LCDSA). The authors combine rigorous experimental characterization with coarse-grained simulations to elucidate a two-stage mechanism (fusion growth followed by coupling) for forming segmented nanowires. This work convincingly establishes LCDSA as a versatile strategy for constructing hierarchical nanostructures. The solvent-dependent tunability of coupling efficiency and the analogy to "living polymerization with termination" are particularly compelling. While the study is well-executed and addresses a knowledge gap in supramolecular self-assembly, several issues need to be addressed before publication in Nature Communications.

1. When feeding the copolymers, the authors used DMF as a good solvent for dissolving the copolymers, while the good solvent used for preparing the seeds was THF. Could DMF affect the growth and coupling of the seed micelles? The role of DMF (as a cosolvent) requires clarification.
2. In this work, the chain rearrangement of PBLG blocks has been shown to be critical to the growth and coupling of cylindrical micelles. The authors stated, "full alignment requires extended time due to the twisting rearrangement of the coupled LC blocks" (Page 11). This statement requires additional simulation or experimental support, specifically regarding the extended time required for chain rearrangement.
3. When studying the effect of methanol content, the authors state that increasing methanol content enhanced the interaction between the PBLG blocks and reduced their rearrangement capability. Can the simulations provide further valuable information supporting this statement?
4. The authors investigated the effect of different solvent compositions on the growth and coupling of seed micelles. The initial good solvents include THF and dioxane, while the selective solvents include methanol and ethanol. Apart from the presented results for various solvent combinations, what are the self-assembly, growth, or coupling behaviors in the dioxane/ethanol system?
5. In Figure 1, the authors should label seed and grown segments directly in TEM images (e.g., Fig. 1d-e) for clarity.
6. If possible, providing videos of simulation trajectories as Supplementary Movies would help visualize growth and coupling behaviors.
7. In Section 1.4 of SI, the authors used "CWC" to describe the critical methanol content is inappropriate. It should be modified as "critical methanol content (CMC)".

Reviewer #2

(Remarks to the Author)

The authors synthesized PBLG-b-PNIPAM block copolymers and prepared segmented polymer nanowires by modulating the fusion-growth process and the end-to-end coupling of cylindrical micelles. Unfortunately, end-to-end coupling for hierarchical self-assembly of nanowires has already been reported, and it is fairly clear that block copolymer micelles can form nanowires via end-to-end fusion even in amorphous, non-liquid-crystalline systems. Consequently, attributing end-to-end coupling to "liquid-crystalline properties promoting flow-mediated fusion" does not represent a unique advance. This reviewer is not able to recommend the publication of this manuscript in Nature Communications.

Below are a couple of detailed comments:

1. In Figure 2c, two peaks are observed at early periods. The authors attribute the smaller peak to monomers/seeds and the

larger peak to aggregates that form over time. However, DLS can also yield multiple peaks simply because the resulting aggregates possess a broad size distribution. To corroborate their interpretation, the authors should provide corresponding TEM or SEM images recorded at the same time points.

2. The authors argue that the liquid-crystalline nature of the core accelerates flow-mediated fusion and thus enables end-to-end coupling. Are there, however, any reports of end-to-end coupling in purely amorphous or non-liquid-crystalline systems that would refute this claimed role of the mesogenic ordering?

3. The authors employed the same material (PBLG₂₃₇-b-PNIPM₁₁₈) to construct both SEGMENTED and GROWN objects. How, then, did they distinguish the two domains?

4. In Figure 1b, d–e, and g–h, why does the apparent width of the SEGMENTED regions vary markedly?

5. Furthermore, why do the SEGMENTED lengths reported in Figure 1i differ from those in Figure S4c? A clear and consistent explanation from the authors is required.

Reviewer #3

(Remarks to the Author)

The manuscript reports on the end-to-end coupling of cylindrical micelles formed by PBLG-b-PNIPAM block copolymers, giving rise to the creation of nanowire structures. The micelles are composed of two segments; one is crystalline core, and the liquid crystal blocks ends featured by disordered inner segment structures. The fusion between liquid crystal blocks accompanied by inner chain rearrangement process drove the end-to-end couplings. The author conducted various experiments including microscopic (TEM and AFM) observations, DLS, XRD studies, and simulations, which supported authors view. The scientific rational and conducted experiments is rigorous and of good quality. However, some experimental results are unclear due to an absence of detail explanation. Thus, I suggest to consider the manuscript for publication in Nature Communications after a major revision in which the following points have to be appropriately addressed in detail.

1. Firstly, while I understand that the end-to-end coupling approach is unique, I am not convinced that it is the only possible route to obtaining the observed nanowire structures. The manuscript does not sufficiently explain the novelty of the resulting polymeric architecture. For example, I suspect that even if the seed solutions were added in a stepwise manner, similar nanowire structures might be obtained. The authors should clarify this point. Likewise, the scope and limitations of the present strategy—as well as its generality—should be explicitly discussed. Along this line, I find no clear explanation of why the PBLG-b-PNIPAM block copolymer was chosen for this study. Is this specific structure (or its segmental distribution) indispensable for enabling end-to-end coupling?

2. For the resulting nanowires, more detailed statistical analyses of the internal structure are required to support the proposed mechanism. While the time-dependent DLS data provide information about the hydrodynamic radius (i.e., overall length), detailed structural insights into the internal organization are lacking. To this end, I feel the analysis of TEM and AFM images is not managed well. A statistically meaningful number of nanowires should be measured both before and after incubation, and each LC segment length should be analyzed in much detail. Such analyses would provide quantitative data on how many seeds (or unimers) were consumed and converted during the elongation process. These quantitative data would confirm that no unreacted seeds' ends remained prior to adding polymers, which should be critical to work out the supramolecular end-to-end mechanism.

3. The discussion of several important aspects is overly brief. For instance, the solvent effect investigated in the final part of the manuscript is mentioned only superficially. I can reasonably infer that solvation at the micelle edges influences reactivity, likely due to variations in mobility within the LC domain. However, a more detailed molecular-level discussion seems to be lacking. Which segments in the copolymer are preferentially solvated by the selected solvents? What is induced inside upon changing the polarity (solubility) parameters? In addition, incubation temperature would also influence on the edge reactivity. How the authors optimized these parameters?

4. The terms such as “crystalline and liquid-crystalline” used for distinction between two parts in the micelle are somewhat ambiguous for non-experts. Related to this, the provided schematic illustration may confuse readers who are not familiar with polymer assemblies. After reading, I understand that the green and purple domains differ in their assembled structures (degree of crystallinity) but are chemically identical. The authors should make this distinction clearer to avoid misinterpretation.

Version 1:

Reviewer comments:

Reviewer #1

(Remarks to the Author)

The authors have fully addressed all my concerns. The manuscript is ready for publication now.

Reviewer #2

(Remarks to the Author)

It is worth noting again that end-to-end coupling of cylindrical micelles has already been reported in living CDSA systems (e.g. liquid crystalline block copolymer: *Macromolecules* 2021, 54, 6845–6853; *Nature Communications*, 2024, 15, 2968; crystalline-coil block copolymer: *J. Am. Chem. Soc.* 2011, 133, 11220-11230), and investigations on end-to-end supramolecular assembly of cylindrical micelles have been more intensively published. The authors only (partially) addressed the technical issues, but failed to elevate the basic novelty of this work. Thus, this reviewer strongly feel that it is not suitable for a high-tier journal like *Nature Communications*.

Reviewer #3

(Remarks to the Author)

The authors have done an excellent job of addressing all my comments, as well as those from the other reviewers, in a detailed and satisfactory manner. They have made appropriate changes to both the main manuscript and the Supplementary Information accordingly. In addition, they have carried out a series of new experiments and theoretical simulations, all of which have contributed to improving and clarifying the content of the paper. I sincerely appreciate the authors' careful revisions. In my opinion, all the points raised by the reviewers have been clearly addressed, and the revised manuscript is now suitable for publication in *Nature Communications*.

I would like to congratulate the authors on their conceptually insightful and well-executed work.

Responses to Reviewer's comments

We thank the reviewers for their careful review of our manuscript and for providing comments and suggestions to improve the quality of the manuscript. Our point-by-point responses to these comments are given below.

Reviewer #1:

This manuscript from Gao et al. presents a significant advancement in supramolecular polymer science by demonstrating controlled end-to-end coupling of cylindrical micelles via liquid-crystallization-driven self-assembly (LCDSA). The authors combine rigorous experimental characterization with coarse-grained simulations to elucidate a two-stage mechanism (fusion growth followed by coupling) for forming segmented nanowires. This work convincingly establishes LCDSA as a versatile strategy for hierarchical nanostructures. The solvent-dependent tunability of coupling efficiency and the analogy to "living polymerization with termination" are particularly compelling. While the study is well-executed and addresses a knowledge gap in supramolecular self-assembly, several issues need to be addressed before it to be published in Nature Communications.

Comment 1.1 *When feeding the copolymers, the authors used DMF as a good solvent for dissolving the copolymers, while the good solvent used for preparing the seeds was THF. Will DMF affect the growth and coupling of the seed micelles? The role of DMF (as a cosolvent) should be clarified.*

Response: DMF, relative to THF, has a better solvation power for PBLG-*b*-PNIPAM. We choose THF for seed preparation because DMF leads to irregular aggregates. For seeded growth, we need to prepare a highly concentrated polymer solution to minimize the amount of good solvents in the growth system, so DMF is used.

To test whether DMF influences micelle growth and end-to-end coupling, we added copolymers dissolved in different amounts of DMF into seed solutions. We monitored the assemblies for over 12 h and 72 h. Despite differences in DMF content, all samples exhibited similar behavior: seeds grew into cylindrical micelles at 12 h. They later coupled into segmented nanowires, with each nanowire containing more than three seeds. These results show that small amounts of DMF do not affect either micelle growth or coupling; instead, it dissolves well the added copolymer during feeding.

This information has been incorporated into the revised manuscript (see Line 11-15 of Page 5 in the main text, and Section 1.2 of the Supporting Information, SI).

Comment 1.2 In this work, the chain rearrangement of PBLG blocks has been demonstrated to be critical to the growth and coupling of cylindrical micelles. The authors mentioned "full alignment requires extended time due to the twisting rearrangement of the coupled LC blocks" (Page 11). This statement requires additional simulation or experimental support.

Response: We thank the reviewer for this helpful suggestion. Our original statement states that chain rearrangement during end-to-end coupling takes significantly longer than that during fusion growth. To support this, we compared the simulation times required for the two processes. The results indicate that the fusion growth between a seed and a small aggregate takes ca. 1500τ for the chains to rearrange (**Fig. R1a**). In contrast, the coupling of two grown cylindrical micelles takes about 5500τ (**Fig. R1b**). This confirms that chain rearrangement in the coupling step is indeed much slower than in fusion growth.

Fig. R1. The consumed simulation time for chain rearrangements: (a) fusion growth of seed and small aggregate, (b) coupling of two contacted grown cylindrical micelles, where τ is the unit of simulation time.

We have incorporated the results and discussion into the revised manuscript (see Line 11-13 of Page 12 in the main text, and Section 2.3 of the SI).

Comment 1.3 When studying the effect of methanol content, the authors mentioned that the increase in methanol content enhanced the interaction of the PBLG blocks and reduced the rearrangement ability. Can the simulations provide more valuable information for this statement?

Response: According to the reviewer's suggestion, we increased the interaction parameter ϵ_{RR} in the simulations, which corresponds to the enhanced interaction of the PBLG blocks at high methanol content in the experiments.

When the ϵ_{RR} is 2.7ϵ , the fusion growth and end-to-end coupling of cylindrical micelles occur, and the ordered packing manner of chains can be formed in the grown parts and the coupled parts. When the ϵ_{RR} is increased to 2.8ϵ (corresponding to higher methanol content), the fusion growth and end-to-end coupling can still be observed. However, the chain packing manners in the formed nanowires are relatively less disordered. **Fig. R2** shows the comparison of the variations in the cosine of orientation angles, φ , along the long axis of coupled micelles formed at 2.7ϵ and 2.8ϵ , respectively. At the interaction strength of 2.8ϵ , the LC chains within the ends of the grown micelles have an imperfect cholesteric LC arrangement, and the chains can not well rearrange themselves after coupling (see **Fig. R2b**). This result indicates that the enhancement of the interaction parameter ϵ_{RR} (higher methanol content) makes the chain rearrangement more difficult and limits the fusion growth and coupling of micelles. This is consistent with the experimental observation.

Fig. R2. Variations in the cosine of φ along the X-axis of the cylindrical micelles obtained at the ϵ_{RR} of (a) 2.7ϵ and (b) 2.8ϵ . Only the two ends of the grown micelles and their coupled parts are presented in the figures.

The above simulation results and discussion provide more information to support our statement regarding chain rearrangement ability at high methanol content, which has been incorporated into the revised manuscript (see Line 10-13 of Page 13 in the main text, and Section 2.4 of the SI).

Comment 1.4 The authors investigated the effect of different solvent components on the growth and coupling of seed micelles. The initial good solvents include THF and Diox, while the selective solvents

include MeOH and EtOH. Apart from the presented results of various solvent components, how about the self-assembly, growth, or coupling behaviors in the Diox/EtOH system?

Response: As the reviewer suggested, we prepared seed micelles under Diox/EtOH conditions and examined the growth and coupling behaviors of micelles in this solvent condition, where the solvent ratio, polymer concentration, and temperatures remain unchanged.

At the 45.5/54.5 of Diox/EtOH in volume, the obtained seeds have lengths of ~550 nm (**Fig. R3a**). Subsequently, DMF solutions of the block copolymer (2.0 g/L, 40 μ L) were added to these seed solutions for growth. After incubation for 48 hours, the segmented nanowires can be observed in the Diox/EtOH/DMF system (**Fig. R3b**). We tracked the variation of the number of seeds (N^{SEED}) per nanowire over time, and the results are shown in **Fig. R3c**. The N^{SEED} in the Diox/EtOH/DMF system reaches 3.8 (see the solid curve in Fig. R3c). In addition, the total length of grown parts (L^{GROWN}) reaches ~1800 nm. It was found that the micelle growth and coupling in the Diox/EtOH/DMF system are more favorable than those in the Diox/MeOH/DMF system. Overall, the Diox/EtOH/DMF system supports growth and coupling even more favorably than the Diox/MeOH/DMF system, confirming that solvent composition regulates chain rearrangement and thus the final nanowire morphology.

Fig. R3. (a) TEM image of the seed micelles prepared in Diox/EtOH mixture solution. (b) TEM image of the segmented nanowires prepared in Diox/EtOH/DMF mixture solution after aging at 30 °C for 48 hours. Scale bars: 1 μ m. (c) Temporal variation of the number of seeds (N^{SEED}) for the Diox/EtOH/DMF system, and the corresponding temporal variations for the other solvent conditions are also plotted for comparison.

These contents have been incorporated into the revised manuscript (see Fig. 5, Line 2-6 of Page 15 in the main text, and Section 1.13 of the SI).

Comment 1.5 *In Figure 1, labeling seed and grown segments directly in TEM images (e.g., Fig. 1d-e) is suggested for clarity.*

Response:

According to the reviewer's suggestion, the seed and grown segments have been clearly labeled in the revised manuscript (see Fig. 1e).

Comment 1.6 *If available, provide videos of simulation trajectories as Supplementary Movies to visualize growth and coupling behaviors.*

Response:

We made a video of simulation trajectories as suggested. We provided it as Supplementary Movies to visualize growth and coupling behaviors (see the separate file named "Supplementary Movies" in the revised version).

Comment 1.7 *In Section 1.4 of SI, the authors used "CWC" to describe the critical methanol content is inappropriate. It should be modified as "critical methanol content (CMC)".*

Response:

We recognized that the critical methanol content and its abbreviation in the previous version were incorrectly written as "critical water content (CWC)." In the revised manuscript, the term has been modified to "critical methanol content (CMC)" (see Section 1.4 of the SI).

Reviewer #2

The authors synthesized PBLG-*b*-PNIPAM block copolymers and prepared segmented polymer nanowires by modulating the fusion-growth process and the end-to-end coupling of cylindrical micelles. Unfortunately, end-to-end coupling for hierarchical self-assembly of nanowires has already been reported, and it is fairly clear that block copolymer micelles can form nanowires via end-to-end fusion even in amorphous, non-liquid-crystalline systems. Consequently, attributing end-to-end coupling to “liquid-crystalline properties promoting flow-mediated fusion” does not represent a unique advance. This reviewer is not able to recommend the publication of this manuscript in *Nature Communications*.

Comment 2.1 In Figure 2c, two peaks are observed at early periods. The authors attribute the smaller peak to monomers/seeds and the larger peak to aggregates that form over time. However, DLS can also yield multiple peaks simply because the resulting aggregates possess a broad size distribution. To corroborate their interpretation, the authors should provide corresponding TEM or SEM images recorded at the same time points.

Response: According to the reviewer’s suggestion, we supplemented the TEM images corresponding to various time points in the DLS curves (**Fig. R4a**, i.e., Fig. 2c mentioned by the reviewer).

Fig. R4. (a) Temporal evolution of the R_h of the seed solution containing added copolymer unimers. (b-f) TEM images corresponding to various time points in the DLS curves of (b) 0 hours, (c) 6 hours, (d) 24 hours, (e) 48 hours, and (f) 72 hours, respectively. Scale bars: 1 μm .

Small aggregates are visible in **Fig. R4b**, matching the small peak in the DLS data. As incubation continues, this peak gradually weakens and disappears (red and blue curves in Fig. R4a). TEM images confirm that these aggregates fuse onto the ends of the seed micelles during growth (**Fig. R4c**, 6 h; **Fig. R4d**, 24 h). After 24 hours, end-to-end coupling becomes dominant, producing segmented nanowires (**Fig. R4e**, 48 h; **Fig. R4f**, 72 h). At this stage, R_h increases, and the DLS peaks broaden (green and purple curves). Overall, the TEM observations are fully consistent with the DLS results.

The above results and discussion have been incorporated into the revised manuscript (see Line 21-22 of Page 8 and Line 1 of Page 9 in the main text, and Section 1.11 of the Supporting Information, SI).

Comment 2.2 *The authors argue that the liquid-crystalline nature of the core accelerates flow-mediated fusion and thus enables end-to-end coupling. Are there, however, any reports of end-to-end coupling in purely amorphous or non-liquid-crystalline systems that would refute this claimed role of the mesogenic ordering?*

Response: It is correct that end-to-end coupling is not exclusive to liquid-crystalline systems. Examples exist in amorphous and crystallizable polymers through hydrophobic interactions or crystallization-driven aggregation (Macromolecules 2020, 53, 8581; Polym. Chem. 2023, 14, 2987; Nat. Commun. 2024, 15, 2968).

However, the significance and novelty of our work do not lie in simply observing end-to-end coupling. Instead, our contribution is to uncover *how* the liquid-crystalline (LC) nature of the PBLG core governs the fusion and coupling processes and, critically, how this LC-driven chemistry enables *fine-tuning* of the coupling behavior.

In our system, both fusion growth and end-to-end coupling require LC chain rearrangement, and the coupling step is particularly sensitive to the efficiency of this rearrangement. This provides a robust and previously underappreciated handle: by modulating solvent composition, we can directly influence LC ordering, thereby controlling growth, coupling dynamics, and ultimately the hierarchical structure of the resulting nanowires.

Thus, the key advance of this work is not the occurrence of coupling itself, but the mechanistic insight that liquid-crystalline chemistry offers a precise, tunable pathway to engineer segmented nanowires — an opportunity not available in amorphous or non-LC systems. The above discussion has been given in the revised manuscript (see Line 21-22 of Page 16 and Line 1-3 of Page 17 in the main text).

Comment 2.3 *The authors employed the same material (PBLG₂₃₇-b-PNIPM₁₁₈) to construct both SEGMENTED and GROWN objects. How, then, did they distinguish the two domains?*

Response: From the high-magnification TEM image (see **Fig. R5**), it is clearly observed that there are significant differences in the widths of the two regions. We analyzed more than 200 TEM images using the Image-Pro Plus software and automatically identified the seed and grown segments per nanowire based on the difference in width.

As shown in the image, the thin segments of ~15 nm and the broad segments of ~30 nm alternate along the nanowires. It indicates that growth occurs at the two ends of the seeds, and the LC blocks are organized into a narrower micelle core in the growing segment. Thinner grown segments are due to the lower ordering of LC blocks in different solvent conditions.

Fig. R5. High-magnification TEM image of segmented nanowires. Scale bars: 1 μm .

For clarity, we labeled the seed and grown segments directly in TEM images and provided the identification details in the revised manuscript (see Fig. 1e, Line 22-23 of Page 5, Line 2-4 of Page 6 in the main text, and Section 1.6 of the SI)

Comment 2.4 *In Figure 1b, d–e, and g–h, why does the apparent width of the SEGMENTED regions vary markedly?*

Response: We thank the reviewer for pointing this out. The scale bar in Fig. 1e (the enlarged TEM image) was indeed inconsistent with that in Fig. 1f. In the revised manuscript, we have corrected the scale bar in Fig. 1e and rechecked all figures to ensure consistency (see the updated Fig. 1e and caption).

Comment 2.5 Furthermore, why do the SEGMENTED lengths reported in Figure 1i differ from those in Figure S4c? A clear and consistent explanation from the authors is required.

Response: We appreciate the reviewer's attention to this point. The SEGMENTED lengths in Fig. 1i and Fig. S4c in the previous version appear different because they come from *different experimental batches* and serve *different purposes*.

Fig. 1i reports the **final** SEGMENTED lengths measured after 72 hours at various unimer-to-seed ratios, while Fig. S4c shows the **time-dependent evolution** of SEGMENTED lengths during the growth process.

When we compare only the data collected under the *same* conditions (72 hours and identical unimer-to-seed ratios), the values in Fig. 1i and Fig. S4c agree very well. For example, at ratios of 2, 3, and 5, the final lengths are $2.63 \pm 0.13 \mu\text{m}$, $3.12 \pm 0.20 \mu\text{m}$, and $4.37 \pm 0.22 \mu\text{m}$ in Fig. 1i, which match the corresponding values of $2.60 \pm 0.13 \mu\text{m}$, $3.13 \pm 0.18 \mu\text{m}$, and $4.28 \pm 0.12 \mu\text{m}$ in Fig. S4c. The small differences fall within normal statistical variations.

We have added the numerical comparison and a brief explanation of this consistency in the revised Supporting Information (see Section 1.8 of the SI).

Reviewer #3

The manuscript reports on the end-to-end coupling of cylindrical micelles formed by PBLG-b-PNIPAM block copolymers, giving rise to the creation of nanowire structures. The micelles are composed of two segments; one is crystalline core, and the liquid crystal blocks ends featured by disordered inner segment structures. The fusion between liquid crystal blocks accompanied by inner chain rearrangement process drove the end-to-end couplings. The author conducted various experiments including microscopic (TEM and AFM) observations, DLS, XRD studies, and simulations, which supported authors view. The scientific rational and conducted experiments is rigorous and of good quality. However, some experimental results are unclear due to an absence of detail explanation. Thus, I suggest to consider the manuscript for publication in Nature Communications after a major revision in which the following points have to be appropriately addressed in detail.

Comment 3.1 *Firstly, while I understand that the end-to-end coupling approach is unique, I am not convinced that it is the only possible route to obtaining the observed nanowire structures. The manuscript does not sufficiently explain the novelty of the resulting polymeric architecture. For example, I suspect that even if the seed solutions were added in a stepwise manner, similar nanowire structures might be obtained. The authors should clarify this point.*

Response: We thank the reviewer for this critical comment. We want to clarify that the novelty of our work does not lie in producing nanowires per se, but in creating a *hierarchically segmented* nanowire architecture through a *controllable* end-to-end coupling mechanism driven by liquid-crystalline (LC) chain rearrangement.

The segmented nanowires reported here feature alternating thin and thick segments along their long axis — an architecture fundamentally different from previously reported co-micelles or wormlike micelles (Nat. Commun. 2024, 15, 2968; Macromolecules 2020, 53, 8581; Polym. Chem. 2023, 14, 2987). Their formation relies on the liquid-crystallization-driven self-assembly (LCDSA) mechanism: growth first proceeds through the fusion of small aggregates, followed by LC-mediated end-to-end coupling, which requires substantial chain rearrangement. This LC-based chain rearrangement is what enables fine control over coupling behavior; by tuning solvent composition, we can modulate this rearrangement and thus precisely adjust the hierarchy of the segmented nanowires.

To address the reviewer's question regarding stepwise seed addition, we performed an additional control experiment. Adding more seed solution after 24 hours still produces segmented nanowires, but

increases the average number of seeds per nanowire from ~ 3.3 to ~ 4.5 (**Fig. R6**). This demonstrates that multiple feedings can adjust the hierarchical structure, yet the underlying assembly still proceeds through the exact LC-driven end-to-end coupling mechanism.

In summary, while segmented structures could, in principle, form under different feeding strategies, the key novelty of our work lies in uncovering and exploiting the LC-driven chemical mechanism that enables the *controllable and tunable* formation of hierarchical segmented nanowires — an advance not available in amorphous or non-LC systems.

Fig. R6. Schematic illustration and representative morphology of segmented nanowires prepared by stepwise addition of seed. Scale bars: $1\mu\text{m}$.

To provide a profound understanding of end-to-end coupling, we have incorporated the above results and discussion into the revised version (see Line 3-13 of Page 17 in the main text, and Section 1.9 of the Supporting Information, SI).

Comment 3.2 *Likewise, the scope and limitations of the present strategy - as well as its generality- should be explicitly discussed. Along this line, I find no clear explanation of why the PBLG-*b*-PNIPAM block copolymer was chosen for this study. Is this specific structure (or its segmental distribution) indispensable for enabling end-to-end coupling?*

Response: We appreciate the reviewer’s insightful comments. Below, we clarify both the generality of our strategy and the rationale for choosing the PBLG-*b*-PNIPAM system.

Our work reveals a **liquid-crystallization-driven self-assembly (LCDSA) mechanism** in which micelles first grow through fusion and then undergo LC-mediated end-to-end coupling. A key insight is that **LC chain rearrangement — not simply micelle collision — governs whether controlled coupling can occur**. By tuning solvent composition or temperature, we can regulate this rearrangement ability and thus adjust the hierarchical segmented structures. This mechanistic understanding, rather than the mere

existence of coupling, is the main novelty and significance of the study.

To assess generality, we complemented experiments with theoretical simulations. These simulations revealed general design rules for achieving controlled LC-driven coupling:

1. LC cores should rearrange more slowly during coupling than during fusion, requiring extended twisting and alignment.
2. The solubility parameters of the LC rod block and coil block should be sufficiently close to allow this rearrangement.
3. A moderate block ratio (e.g., ~2:1 for PBLG₂₃₇-*b*-PNIPAM₁₁₈) is needed to balance packing frustration and flexibility.
4. Optimized solvent conditions should allow tunable LC ordering.

When these conditions are met, the hierarchical segmented nanowires can form, demonstrating the **mechanism's generality**.

We also tested other LC-containing block copolymers. For example, PBLG₂₂₆-*b*-PEG₁₁₂ failed to produce segmented nanowires in either THF/MeOH/DMF or Diox/MeOH/DMF, yielding only irregular fibers or cylinders (**Fig. R7**). This is consistent with the design rules: PEG is far more hydrophilic, and its solubility mismatch with PBLG prevents the necessary LC rearrangement.

Fig. R7. (a,c) TEM images of seed micelles prepared by PBLG₂₂₆-*b*-PEG₁₁₂ copolymers in (a) THF/MeOH and (c) Diox/MeOH solutions. (b,d) TEM images of aggregates after feeding PBLG-*b*-PEG copolymers and aging at 30 °C for 48 hours in (b) THF/MeOH/DMF and (d) Diox/MeOH/DMF solutions. Scale bars: 1 μm .

In contrast, PBLG-*b*-PNIPAM satisfies all requirements. PBLG and PNIPAM have closely matched Hansen solubility parameters, and the THF/MeOH/DMF solvent combination provides the right balance for LC ordering and rearrangement. Therefore, the choice of PBLG₂₃₇-*b*-PNIPAM₁₁₈ is not arbitrary — it is essential for enabling the controllable LC-driven coupling process.

These clarifications and supporting data have been incorporated into the revised manuscript (see Line 4-5 of Page 5 and Line 1-11 of Page 18 in the main text, and Section 1.7 of the SI).

Comment 3.3 *For the resulting nanowires, more detailed statistical analyses of the internal structure are required to support the proposed mechanism. While the time-dependent DLS data provide information about the hydrodynamic radius (i.e., overall length), detailed structural insights into the internal organization are lacking. To this end, I feel the analysis of TEM and AFM images is not managed well. A statistically meaningful number of nanowires should be measured both before and after incubation, and each LC segment length should be analyzed in much detail. Such analyses would provide quantitative data on how many seeds (or unimers) were consumed and converted during the elongation process. These quantitative data would confirm that no unreacted seeds' ends remained prior to adding polymers, which should be critical to work out the supramolecular end-to-end mechanism.*

Response: We thank the reviewer for this important suggestion. In response, we have performed a more systematic and quantitative analysis of TEM images throughout the entire incubation period to support the proposed LCDSA mechanism directly.

First, we supplemented TEM images at all key time points corresponding to the DLS measurements (**Fig. R8**). At 0 hours, small aggregates are clearly visible, matching the small DLS peak. As incubation proceeds, these aggregates progressively disappear, and TEM images show that they are consumed through fusion growth at the open seed ends (6 h and 24 h). After 24 hours, end-to-end coupling becomes dominant, and segmented nanowires appear (48 h and 72 h), consistent with the broadening R_h distribution in DLS. Thus, the structural evolution observed by TEM and DLS is fully correlated.

To provide statistically meaningful evidence, we quantified the temporal distributions of seed number (N^{SEED}) and grown length (L^{GROWN}) for the 54.0 vol% MeOH condition (**Fig. R9**). Early in incubation (<6 h), most structures contain only 1–2 seeds, reflecting fusion growth. By 24 h, N^{SEED} increases to 3–4, and by 48 h it stabilizes at ~4, indicating that **nearly all initial seeds have participated in coupling**, with no

unreacted ends remaining. L^{GROWN} also increases rapidly within 6 h and narrows with time, confirming a transition from growth to controlled coupling.

Fig. R8. (a) Temporal evolution of the R_h of the seed solution containing added copolymer unimers. (b-f) TEM images corresponding to various time points in the DLS curves of (b) 0 hours, (c) 6 hours, (d) 24 hours, (e) 48 hours, and (f) 72 hours, respectively. Scale bars: 1 μm .

Fig. R9. Temporal variations of (a) the seed number N^{SEED} and (b) the L^{GROWN} for the segmented nanowires formed at methanol contents of 54.0 vol%.

These analyses directly support the novelty of our mechanism: the hierarchical segmented nanowires arise *not* from random aggregation, but from a time-resolved LC-driven sequence of fusion followed by controlled end-to-end coupling, enabled by LC rearrangement at partially open ends. The absence of unreacted seeds at later stages further confirms that the coupling process is systematic and governed by the LC ordering principles revealed in this work.

All additional analyses and discussions have been included in the revised manuscript (see Line 21-22 of Page 8 and Line 1-3 of Page 9 in the main text, and Section 1.11 of the SI).

Comment 3.4 *The discussion of several important aspects is overly brief. For instance, the solvent effect investigated in the final part of the manuscript is mentioned only superficially. I can reasonably infer that solvation at the micelle edges influences reactivity, likely due to variations in mobility within the LC domain. However, a more detailed molecular-level discussion seems to be lacking. Which segments in the copolymer are preferentially solvated by the selected solvents? What is induced inside upon changing the polarity (solubility) parameters? In addition, incubation temperature would also influence on the edge reactivity. How the authors optimized these parameters?*

Response: We thank the reviewer for the insightful comments. We agree that a deeper discussion of solvent effects and incubation temperature is essential for understanding the molecular basis of micelle growth and end-to-end coupling. These considerations were indeed integrated into our study, and we now provide a more detailed explanation to clarify the underlying mechanism and highlight the novelty of our work.

1) Molecular-level solvation effects

The solvent-dependent behavior originates from the liquid-crystalline (LC) ordering of the PBLG core-forming blocks. **Table R1** summarizes the Hansen solubility parameters of PBLG, PNIPAM, and the solvents used. Because the end-to-end reactivity is dictated by how readily PBLG chains can rearrange at micelle edges, the solvation environment plays a decisive role.

In selective alcohols, methanol ($\delta = 30.20 \text{ MPa}^{1/2}$) solvates PBLG more strongly than ethanol ($\delta = 26.60 \text{ MPa}^{1/2}$). The weaker solvation of PBLG in ethanol introduces a mild solvophobic drive that facilitates chain mobility and LC reorganization. Consequently, growth and end-to-end coupling occur more efficiently in the THF/EtOH/DMF mixture, consistent with our experimental results (see Fig. R3c and Fig. 5 in the revised manuscript). By explicitly incorporating these solubility considerations, we clarify how solvent

identity controls segment rearrangement and thus the kinetics of fusion and coupling. These analyses strengthen the molecular-level foundation of the proposed mechanism.

Table R1. Hansen solubility parameters of PBLG, PNIPAM, and solvents

Sample	Hansen solubility parameters δ / MPa ^{1/2}
PBLG	23.40
PNIPAM	23.50
DMF	24.90
THF	19.50
Diox	20.47
MeOH	30.20
EtOH	26.60

2) Optimization of incubation temperature

Temperature also modulates chain mobility within both the LC core (PBLG) and corona (PNIPAM), thereby regulating edge reactivity.

- At 20 °C, the LC blocks possess insufficient mobility. Chain rearrangement is strongly hindered, and only one or two segments form even after 48 h (**Fig. R10a**).
- At 40 °C, PBLG mobility increases, but PNIPAM chains collapse more tightly around the core, paradoxically restricting PBLG rearrangement and suppressing growth and coupling (**Fig. R10b**).
- At 30 °C, both PBLG and PNIPAM retain adequate mobility. This balanced condition uniquely permits efficient fusion growth followed by controlled end-to-end coupling, as confirmed by the evolution of N^{SEED} (**Fig. R10c**).

These results demonstrate that precise thermal control is critical for enabling the LC-mediated rearrangement required for segment fusion, and they reinforce the central novelty of this work: the coupling behavior is not simply dictated by end-to-end contact. Still, it emerges from tunable LC chemistry at micelle edges.

Fig. R10. TEM images of the aggregates after incubating at (a) 20 °C and (b) 40 °C for 48 hours. (c) Variations of the seed number N^{SEED} for the segmented nanowires formed at various temperatures. Scale bars: 1 μm .

The above results and discussion have been incorporated into the revised manuscript (see Line 1-7 of Page 16 in the main text, and Section 1.13 and 1.14 of the SI).

Comment 3.5 The terms such as “crystalline and liquid-crystalline” used for distinction between two parts in the micelle are somewhat ambiguous for non-experts. Related to this, the provided schematic illustration may confuse readers who are not familiar with polymer assemblies. After reading, I understand that the green and purple domains differ in their assembled structures (degree of crystallinity) but are chemically identical. The authors should make this distinction clearer to avoid misinterpretation.

Response: We thank the reviewer for this suggestion. In the revised manuscript, when describing the differences between the seeds and the grown segments, we emphasize that their chemical composition is the same (both are formed from PBLG-*b*-PNIPAM copolymers), but their structures differ. For clarity, we added a label to the Schematic illustration in Fig. 3i to distinguish between the seeds and the grown segments (see Fig. R11).

Fig. R11. Schematic illustration of (I) Fusion growth and (II) Coupling behaviors via LCDSA, where green and blue objects represent the micelle seed, the grown and coupled segment with different degrees of LC ordering.

The above content has been incorporated into the revised version (see Fig. 3i and its caption).